# Competitiveness and sustainable development of Chinableapple industry

**Yu Sun**[1], **Ye Deng**[1], **Yonghua Lu**[1]*, **Mingyue Li**[2]

**1** College of Economics and Management (Co-operative College)/Qingdao Agricultural University, Qingdao, China, **2** School of Economics and Management/Beijing Forestry University, Beijing, China

* sylyac2006@163.com

**Data Availability Statement:** All relevant data are within the paper and its Supporting Information files.

**Funding:** This research was funded by Shandong Province Modern Agricultural Industrial

## Abstract

Agriculture faces a contradiction between sustainable resource utilization and maintaining market competitiveness. As a major agricultural product, the sustainability and competitiveness of the apple industry have become important topics. This study analyzes the competitiveness of China's apple industry and the factors affecting it. Using 2004 and 2018 data for eight Chinese provinces, principal component analysis and spatial autocorrelation are used to examine competitiveness in terms of five aspects: market, production, technology, organization, and environment. The results indicate that Shandong, Shaanxi, and Gansu were the most competitive during the study period while Hebei, Henan, and Ningxia lagged behind. Regional differences are obvious, with Shandong in particular showing a clear competitive advantage. Although no spatial agglomeration is observed in China's apple industry, with ongoing industrial development, local spatial correlations in the five aspects of competitiveness in the eight provinces have been increasing and gradually stabilizing. This study's findings suggest that improved scientific production, reasonable capital investment, and an established industrial chain are needed to promote local agriculture, economic development, and the central role of the apple industry.

## Introduction

The concept of sustainable development offers a possible solution to the challenges humans face in terms of survival, development, and the pursuit of a high quality of life [1]. In the context of industry, sustainable competitiveness refers to the ability of an industry in one area, compared with the same type industry in other areas [2,3], to accrue and optimize resources and production factors to meet consumers' needs and to survive and develop [4]. In China, the State Council issued a white paper, "Study on China's Population, Resources, Environment, and Development in the Twenty-First Century," which noted that sustainable development is a component of China's long-term plans for social and economic development [5].

Apple production is a major component of Chinese agriculture [6]. Improving the apple industry is important for overall agricultural development in China, and it has practical significance in terms of sustaining farmers' livelihoods [7]. Twenty-four of the 35 varieties of apple are found in China [8]. Apple cultivation in China began in 1913 in Yantai, and after the

Technology System (SDAIT-01-13); the Project of Shandong Provincial Department of Education (M2020111); Qingdao Social Science Planning Project: Research on the Interactive Relationship Between Agricultural Informatization and Agricultural Economic Growth in Qingdao Under the Background of Digital Rural Strategy (QDSKL2101221); Qingdao Double Hundred Research Project: Research on the Coupling Mechanism and Guiding Policies of Digital Investment and Agricultural Modernization in Qingdao (2021-B-15).

**Competing interests:** The authors have declared that no competing interests exist.

"reform and opening up" of the late 1970s, research and technology related to apple cultivation accelerated [9]. In 2019, China's apple-planting area (1.9781 million hm2) and apple production (42.425 million t) accounted for 48% and 54% of the worldwide total, respectively, ranking first in the world. The main apple-producing areas in China are Hebei, Shanxi, Liaoning, Shandong, Henan, Shaanxi, Gansu, and Ningxia [10]. A sound research and development (R&D) system has been established, with numerous technical personnel and researchers engaged in work related to apple production [11]. In 2016, the first World Apple Conference was held in Shaanxi, leading to active international exchange [12].

A mature circulation channel for apples has been formed in China, and considerable progress has been made in related research [13], in areas such as germplasm resources, genetic breeding technology, farming systems, after-picking processing, logistics, and marketing [14]. However, with the rapid development of other fruit industries in recent years, apple industry development has slowed down, and the scale of planting has declined. There are not many apple-planting areas in China, and supply–demand relationships differ between regions [3]. It is important, therefore, to consider production and consumption in different regions to identify regional differences, industrial distribution characteristics, and spatial linkages. Such work can help promote the sustainable competitiveness of China's apple industry.

## Literature review

Sustainable industrial competitiveness has long been a focus of research [15,16]. Diang (1991) was the first to study industrial competitiveness in China [17]. Since then, researchers have investigated areas such as national [18], urban [19], regional [20], enterprise [21,22], and industrial competitiveness in China [23,24]. Research on agricultural competitiveness in China has mainly focused on three areas. The first concerns competitiveness in specific industries, such as sugar [25], corn [26], and fisheries [27]. Studying China's sugar industry, Vacek et al. [25] analyzed the country's competitiveness in terms of production, processing, and consumption; China's sugar industry was found to have deficiencies compared with the international sugar industry, and suggestions for sustainable development were proposed. Kovacevic et al. [26] analyzed the insufficient price competitiveness of corn and the factors affecting the price of corn, and analyzed the key factors affecting the sustainable competitiveness of corn, so as to promote the sustainable development of the industry. Dai et al. [27] used the constant market share model to analyze the market competitiveness of tilapia exported from China and Indonesia to the United States, and further analyzed the factors affecting the export. The study found that changes in market demand had a great impact on fishery breeding, and the market must improve its competitiveness by providing high-quality and high-priced products.

The second area of research focuses on the sustainable competitiveness of the entire agricultural industry in a given region. Chen [28] analyzed the competitiveness of the tea industry in various county-level administrative districts in Fujian Province by measuring the competitiveness of each county-level administrative district and selecting the districts and counties suitable for the planting industry, the characteristics of the tea space layout, and the strong county-level administrative districts with planting advantages. Compared with the county-level administrative districts that have a strong willingness to grow tea, the agglomeration effect, and other important factors, the aspects mentioned above address the competitiveness and spatial structure optimization of the characteristic agricultural industries in county-level administrative districts, and the competitiveness and spatial structure optimization of characteristic agricultural industries in county-level administrative districts. Coordinated development can effectively promote the sustainable competitiveness of agriculture. Maslova [29] made a comparative analysis of the EU's food industry competitiveness. They pointed out that

through agricultural endowments, the level of economic development, and the quality differentiation of exported food products, EU policies and other factors had increased the degree of sustainable food competitiveness. They proposed that sustainable food trade specialization could improve competitiveness or develop a competitive global integrated supply chain management system. Beber [30] proposed strategies for strengthening the publicity about the regional brand image, enhancing awareness of the regional agricultural products, and developing agricultural industrial clusters.

The third area of research on agricultural competitiveness in China investigates the factors affecting sustainable competitiveness in the apple industry. Various important factors have been identified. These include climate resources [6,31], industry agglomeration [32], geographic accessibility [3], prices, labor costs [33], apple quality [34], apple variety [10], orchard construction [31], agricultural technology development [35,36], brands and publicity [4], and the effect of the international environment on exporters. Busdieker and Jesse [35] established a balanced model of time and space in the evaluation of the impact of new technology for the apple industry. Their study compared the transgenic technology and the traditional method and other new methods, such as the biological preparation of microcapsules. Their research also suggests that there are cost benefits from this technology.

The above-mentioned research laid a solid foundation for this article. However, previous discussions on the competitiveness of the agricultural industry, especially the apple industry, used different research methods and perspectives. For example, most researchers typically studied the apple industry competitiveness of a single region. There is a lack of research on the regional specific comparison of the overall apple cultivation in specific countries and the overall global trend. The main innovations of this article are as follows. First, this paper constructs the evaluation index system of apple industry competitiveness based on six dimensions, comprehensively considering the macro and micro scales, to ensure the scientific selection of the three-level index under the six basic criteria. Through the analysis of agricultural economic competitiveness at different levels, the specific problems of each main production area of the apple industry in China were clarified, and reasonable countermeasures and suggestions were developed. Second, we used principal component analysis [37,38] and space exploration analysis [39] as the research methods. Third, we analyzed the changes in the competitiveness of the eight main apple-producing areas and selected the two-year apple-related data in 2004 and 2018 to explore the development and prospects of China's apple industry competitiveness; we analyzed the spatial differentiation characteristics of China's apple industry competitiveness, evaluated the sustainable development potential of the apple industry, which contributes to the development of the apple industry, and provided relevant advice on agricultural progress, with a view to promoting the sustainable development of the apple industry.

## Materials and methods

### Research area and data sources

Apple production in China is widely distributed but is mainly concentrated in Hebei, Shanxi, Liaoning, Shandong, Henan, Shaanxi, Gansu, and Ningxia. Therefore, data from those eight areas are suitable for analyzing the sustainable competitiveness of China's apple industry.

Data for area, yield, and yield per unit area of each province are collected from the provincial statistical yearbooks for 2004 and 2018. The degree of production organization is based on data reported by the National Apple Industry Information Platform. Scientific and technological innovation indicators are collected from the National Agricultural Cost-Benefit Data Compilation. Data on the level of variety are obtained by searching the Internet. Product marketing methods are obtained from the Apple Information Network. Product quality data come from

the selection and appreciation of varieties in online shopping markets, and brand-creation data come from the Apple Information Network. The apple production cost and fertilizer input data of the eight production areas are selected from the National Agricultural Cost-Benefit Data Compilation. Policy support is collected from the agricultural policies of provincial governments related to apples and other fruits during the study period.

## Measurement indicators

The apple-production process is affected by the natural environment, technological levels, agricultural levels, the economic environment, and policy support [40,41]. In this study, guided by Porter's diamond model [42] and other research on industrial competitiveness [43,44], the factors affecting apple industry competitiveness are divided into production factors, technology level, agricultural innovation, brand building, market conditions, and policy support. On that basis, an apple industry competitiveness evaluation index system is established (Table 1).

**Production factors.** In Porter's diamond model [42], production factors are divided into human resources, natural resources, knowledge resources, capital resources, and infrastructure. As important production factors, yield per unit area, planting area, planting scale index,

**Table 1. Evaluation index system for apple industry competitiveness.**

| Target | Guidelines | Indicators | Indicators show |
|---|---|---|---|
| Evaluation index system of apple industryry competitiveness | Factors of production | Yield | Yield per unit area (kg/ha). |
| | | Area | Planted area (ha). |
| | | Planting scale index | Comparison of the proportion of apple acreage in all fruit acreage in the region and the national average level of this proportion (%). |
| | | Output efficiency index | Relative level of the yield rate of apple orchards with the yield rate of all fruits in this region and the comparative relationship with the average level of the yield rate in China (%). |
| | | Composite production index | Geometric mean of the product of planting size index and output efficiency index (%). |
| | Technical level | Degree of organization of production | Histology degree of fruit growers (low to high, 1–5). |
| | | Technological innovation capability | Technical innovation ability (low to high, 1gh, |
| | | Apple variety level | Apple variety level (low to high, 1o hi |
| | Agricultural innovation | Apple marketing | Apple marketing methods (less to more, 1–5). |
| | | Quality evaluation | Apple commodity evaluation (low to high, 1).gh |
| | Branding | Number | Number of well-known brands (individual). |
| | Market conditions | Market share | Annual output/total output in China (%). |
| | | Yield cost coefficient | Regional apple production cost per unit. Output/national apple production cost per unit output (%). |
| | | Output cost coefficient | Production cost per unit value of regional apple/production cost per unit value of national apple (%). |
| | | Regional economic level | GDP by region/national GDP. |
| | | Profit related | Profit per 50 kg apple by region/national average. Profit per 50 kg apple. |
| | Policy support | Policy support | Support strength for apple and other fruit industries (low to high, 1pple |
| | | Publicity | Promotion intensity of apple and other fruit industries (low to high, 1–5). |

Data source: China Statistical Yearbook, compilation of income data for national agricultural products.

output efficiency index, and comprehensive production index are strongly related to competitiveness [44,45]. For this study, the yield per unit area and planting area data are taken from the National Statistical Network, and the planting scale index, output efficiency index, and comprehensive production index are taken from the relevant statistical yearbooks.

**Science and technology level.**  Technological innovation capability refers to the ability of R&D to innovate technologies for the cultivation of different apple varieties in various regions [10], and the degree of production organization is specifically expressed as the organizational degree of fruit farmers in each region [46,47].

**Agricultural innovation.**  Agricultural innovation level is composed of product marketing method, apple variety level, and product quality evaluation [48,49]. Product marketing method refers to the process of promoting and selling apples in a region, apple variety level refers to the number of apple varieties planted in each region, and product quality evaluation refers to online palatability evaluations of the apples from the main production regions.

**Brand building.**  Brand creation is the foundation of market positioning and an embodiment of market competitiveness [50]. The number of brands owned by each region indicates the level of market recognition [51].

**Market conditions.**  Market conditions have a great influence on the apple-trading process. Apples produced in different regions have completely different market shares. The higher the market share, the greater the competitiveness of the region [52]. For fruit production in different regions, the production and output cost coefficients are used, market share and regional economic level are correlated with profit, and the market share of the province is observed.

**Policy support.**  Government policy support is an important reflection of a region's developmental advantage. Local government policies can directly affect farmers' planting processes, the market, and the development of local apple industries [53].

## Methods

### Principal component analysis

Factor analysis is used to extract common factors in the analysis of index data and reduce the dimensionality of many index factors. The basic principle is to extract and classify many indicators with strong correlations and replace a large number of factors with fewer ones to overcome the difficulties caused by large amounts of data, numerous influencing factors, and complex correlations [54,55]. The three functional factors of R&D, production, and sales have many related indicators; thus, the use of factor analysis is reasonable. Principal component analysis (PCA) is further used to reduce dimensionality. It can retain information for all indicators and establish a model for convenient calculation. Thus, this study uses PCA to calculate the related indexes [56,57].

The main steps in factor analysis are as follows: First, the data are standardized, and correlation tests are conducted to check whether the requirements of factor analysis are met. Second, the cumulative contribution rate is observed to determine the principal component factor. Third, the factor loading matrix is solved. Finally, the factor synthesis score is solved [58,59].

In this study, SPSS 22.0 is used to standardize the z-scores for data of different dimensions before PCA. The mean and standard deviation of the data are calculated to achieve standardized data. The formula is new data = (original data − mean)/standard deviation [60,61]. Before PCA, all indicators are divided into five principal components.

The first principal component, F1, includes the output cost coefficient, quality evaluation, output value cost coefficient, number of brands, regional economic level, market share, and profit-related indicators, indicating the apple's share of the fruit market and the corresponding

income. These are also important requirements for the circular development of the apple industry and the rapid improvement of its sustainable competitiveness. These indicators are collectively referred to as "market competitiveness."

The second principal component, F2, includes yield per unit area, planting scale index, output efficiency index, comprehensive production index, and other indicators. It reflects the basic situation of the apple industry in each region, as well as the advantages and disadvantages of apple planting in different provinces. These indicators are referred to as production competitiveness.

The third principal component, F3, includes the science and technology input factor and technology innovation ability index. These reflect industrial science and technology input and industrial innovation input. This is an important reflection of the sustainability of industry competition. These indexes are called technological competitiveness.

The fourth principal component, F4, includes indicators such as the degree of production organization and the level of apple variety. These reflect the quality of farmers in the apple industry and their planting and production ability. They are key to promoting the circular development of the apple industry. These indicators are referred to as organizational competitiveness.

The fifth principal component, F5, includes indicators for policy support and publicity intensity. These pertain to comparisons of external support in different regions and whether governments provide strong support for the development of local apple industries. These indicators are referred to as external environmental competitiveness.

The five principal component scores and composite scores are then calculated and ranked.

## Exploratory spatial data analysis

A spatial weight matrix concerns the degree of "distance" between the objects under study. This can be geographical distance or economic distance. The formula for spatial weight matrix W is as follows:

$$\text{W} = \begin{pmatrix} w_{11} & \cdots & w_{1n} \\ \vdots & \ddots & \vdots \\ w_{n1} & \cdots & w_{nn} \end{pmatrix},$$

where $w_{ij}$ represents the proximity relationship of geographical or economic distance between regions $i$ and $j$, which is usually measured by geographical distance or economic correlation degree. Before the empirical analysis of spatial data, the spatial weight matrix can be obtained by calculating the weight value.

There are various calculation methods for spatial weight matrixes, such as binary adjacency matrix, inverse distance weight matrix, economic weight matrix, and nested matrix. The most common ones are the binary adjacency matrix (rook or queen) and inverse distance weight matrix.

The inverse distance weight method is a spatial distribution method that considers regional correlation among various factors and has been widely used in data processing due to its simple principle and accurate results. The inverse distance weighting matrix is based on the assumption that space effect intensity depends on distance. Except for the main apple-producing units, the effect size of all other evaluation units decreases with increasing distance, which is consistent with the first law of geography stating that "anything is always correlated, and things that are close are more correlated than things that are far away." Meanwhile, a non-sparse matrix can be transformed into a sparse matrix to solve the problem of long running

time caused by excessive number of evaluations. Therefore, the inverse distance weight matrix is used as the basis for spatial autocorrelation analysis.

Using Stata 15.1, the inverse distance weight is used in this study as the benchmark weight. Inverse distance weight is the reciprocal of distance calculated by latitude and longitude. The farther the distance, the smaller the weight, indicating smaller spatial correlation level.

## Global spatial autocorrelation

The formula is as follows.

$$I = \frac{n \sum_{i=1}^{n} \sum_{j=1}^{n} w_{ij}(x_i - \bar{x})(x_j - \bar{x})}{\sum_{i=1}^{n} \sum_{j=1}^{n} w_{ij} \sum_{i=1}^{n} (x_i - \bar{x})^2} = \frac{\sum_{i=1}^{n} \sum_{j \neq i}^{n} w_{ij}(x_i - \bar{x})(x_j - \bar{x})}{S^2 \sum_{i=1}^{n} \sum_{j \neq i}^{n} w_{ij}}$$

Global Moran's I is used to test the spatial autocorrelation of the core variables. The calculation formula of Moran's I is as follows:

$$\bar{x} = \frac{1}{n} \sum_{i=1}^{n} x_i$$

The value range of Moran's I is [−1, 1]. If Moran's I < 0, there is negative autocorrelation; otherwise, there is positive autocorrelation. The greater the absolute value of Moran's I, the greater the autocorrelation.

## Local autocorrelation

The formula is as follows:

$$I_i = Z_i \sum_{i=1}^{n} W_{ij} Z_j$$

By observing the local Moran's I and the scatterplot of Moran's I, the instability of local space can be observed. The four quadrants indicate the relationship between a region and an adjacent region.

## Results

### Overall competitiveness

In the rankings for comprehensive competitiveness, Shandong, Shaanxi, and Gansu all have positive scores (1.4805, 1.0825, and 0.105, respectively), making them the top three provinces.

They are excellent apple-producing areas with strong industrial competitiveness. Meanwhile, the comprehensive scores for Henan, Hebei, and Ningxia are relatively low (−0.4375, −0.4045, and −1.1875, respectively). Although consumer demand has increased in recent years, Henan, Hebei, and Ningxia lag behind because of the low level of planting and lack of science and technology investment.

As shown in Table 2, the rankings of Shanxi and Gansu have been rising, with Shanxi moving up three places for the highest increase. The ranking of Shandong, Shaanxi, and Ningxia hardly changed between 2004 and 2018. Liaoning, Hebei, and Henan fell in the rankings, with Hebei falling the most.

### Competitiveness of different regions

As shown in Table 3, Shanxi Province, in the east of the Loess Plateau, is located in a high-quality industrial belt for apple planting. Over the years, the apple production structure in Shanxi

**Table 2. Overall competitiveness rankings.**

| Provinces | Market competitiveness F1 | | Production competitiveness F2 | | Technical competitiveness F3 | | Organizational competitiveness F4 | | Competitiveness of external environment F5 | | Comprehensive competitiveness F | | Average F | | Average F | Ranking |
|---|---|---|---|---|---|---|---|---|---|---|---|---|---|---|---|---|
| | 2004 | 2018 | 2004 | 2018 | 2004 | 2018 | 2004 | 2018 | 2004 | 2018 | 2004 | 2018 | 2004 | 2018 | | |
| Hebei | 0.009 | 0.112 | −1.295 | −1.016 | −0.018 | −0.112 | −0.385 | 0.236 | −0.577 | −1.127 | −0.288 | −0.391 | −0.426 | −0.383 | −0.4045 | 6 |
| Shanxi | −0.598 | −0.771 | 1.214 | 0.353 | −0.297 | −0.963 | −0.364 | −0.236 | −0.577 | −0.440 | −0.32 | −0.461 | −0.157 | −0.420 | −0.2885 | 4 |
| Liaoning | −0.086 | −0.480 | −0.599 | −0.222 | −0.277 | −0.269 | −0.385 | −0.707 | −0.577 | 0.147 | −0.377 | −0.374 | −0.383 | −0.318 | −0.3505 | 5 |
| Shandong | 2.080 | 1.684 | 0.195 | 0.203 | 2.040 | 2.072 | 1.552 | 1.650 | 1.465 | 1.420 | 1.818 | 1.59 | 1.525 | 1.436 | 1.4805 | 1 |
| Henan | −0.395 | −0.397 | −0.544 | −0.837 | −0.127 | 0.197 | −0.375 | −0.236 | −0.577 | −1.127 | −0.336 | −0.499 | −0.392 | −0.483 | −0.4375 | 7 |
| Shaanxi | 0.710 | 1.030 | 1.615 | 1.438 | 0.623 | −0.144 | 1.552 | 0.943 | 1.465 | 1.420 | 1.196 | 1.143 | 1.194 | 0.971 | 1.0825 | 2 |
| Gansu | −0.570 | 0.251 | 0.141 | 1.262 | −0.537 | 0.426 | −0.385 | 0 | 0.457 | 0.147 | −0.404 | 0.472 | −0.216 | 0.426 | 0.1050 | 3 |
| Ningxia | −1.150 | −1.430 | −0.728 | −1.180 | −1.407 | −1.206 | −1.210 | −1.650 | −1.081 | −0.440 | −1.288 | −1.479 | −1.144 | −1.231 | −1.1875 | 8 |

Province has been adjusted. Based on advanced technology and equipment, both edible and processed apples have fully saturated the market. Meanwhile, in 2010, the government of Gansu Province issued a notice regarding "Industry Development Support Measures" for the local apple industry, which promoted the stable development of the apple industry in Gansu.

Shandong, Ningxia, and Shaanxi showed steady performance during the study period. Shandong, as a traditionally dominant apple-production area, has a temperate monsoon climate and abundant precipitation. It is not too hot in summer or too cold in winter. The soil receives sufficient sunlight, has good drainage, is rich in organic matter, and is slightly acidic to slightly alkaline. Moreover, Shandong's open terrain—a wide range of plain—is conducive to large-scale planting. In 2018, Shandong's government issued the "Promoting the Development of High-Quality Apple Industry Action Plan," reflecting the province's planned investment in science and technology related to apple-production optimization. Shaanxi, meanwhile, has an advantageous natural environment. Its mountains are suited for tree planting, it has low-cost labor, and it has been enhancing its technical capability. Shaanxi ranks first in apple-planting area and has a superior yield. Finally, Ningxia, with its production of concentrated apple juice, has become a production area for the processing and utilization of raw materials. Although its scale is small, the industrial chain is gradually improving. Ningxia ranks low because of its lack of scientific and technological investment, inadequate government policies, and lack of publicity.

Liaoning, Hebei, and Henan fell in the rankings during the study period. Liaoning has an outstanding production scale and technology level, but as a result of weak natural conditions, it cannot form large-scale production or pursue new R&D. Thus, its market competitiveness has gradually weakened. Meanwhile, with its advantageous climate and sufficient rainfall, Hebei has an excellent environment for apple growth. However, it needs to strengthen its investment in science and technology and reform the production structure to obtain strong competitive advantage. Henan, finally, has a hilly area suitable for all kinds of fruit cultivation.

**Table 3. Changes in apple industry competitiveness rankings.**

| | Rising | | Stable | | Falling | |
|---|---|---|---|---|---|---|
| *Provinces* | Shanxi (+3) | Gansu (+1) | Shandong | Ningxia | Liaoning (−1) | Hebei (−2) |
| | | | Shanxi | | Henan (−1) | |

Data source: China Statistical Yearbook, compilation of income data of national agricultural products.

Its largest apple-planting area, Luoning County, is located in the high-quality apple industrial belt of the Loess Plateau. However, because of the limited planting area and resources in Henan, the scale of production has gradually fallen, which is the main reason for its low ranking for comprehensive competitiveness.

## Spatial analysis of apple industrytiveness.le indust

**Global spatial autocorrelation.**    The index test evaluates the significance of the index by calculating Moran's I values, z-scores, and p-values. E(I) is the expected value, SD(I) is the standard deviation, the z-score is a multiple of the SD, and the p-value approximates the area based on the curve of the known distribution, representing probability. Moran's I > 0 indicates positive spatial correlation; the larger the value, the more obvious the spatial correlation. Moran's I < 0 indicates negative spatial correlation; the smaller the value, the greater the spatial difference. Moran's I = 0 indicates random space.

From the results of each comprehensive index, Moran's I basically shows a negative value. The p-values of all indicators are greater than 0.05, indicating that there is no global spatial correlation. In addition, the competitiveness relationship of the apple industry in China may be in a cross-distribution situation in regions, or it may be that only a few regions have spatial correlation. Since the global Moran's I does not show characteristics of spatial dependence and spatial aggregation, the next step is to conduct a spatial autocorrelation test for apple industry competitiveness in local regions. By setting the distance weight, the Moran's I of each variable is obtained as shown in Table 4.

**Local spatial autocorrelation.**    In the scatterplot of Moran's I variables, the X-axis represents the value of the main production area itself, the Y-axis is the value affected, and the oblique line is the trend line. The closer the scatter point is to the origin, the closer its value is to 0, and it is not affected by the surrounding area. The H-h region in the first quadrant indicates that a high-competitive region is surrounded by a high-competitive region. The l-H region in the second quadrant indicates that the low-competitive region is surrounded by a high-competitive region. The l-L region in the third quadrant indicates that a low-competitive region is surrounded by a low-competitive region. The h-L region in the fourth quadrant indicates that a high-competitive region is surrounded by a low-competitive region. Based on the figure, we can analyze the relationships and correlation changes in the industrial competitiveness of the eight main apple-producing areas in the surrounding areas.

**Table 4.  Moran4ed as shown in Table 4.4.ensive index, Morhina.**

| Name | F1 Market competitiveness | | F2 Production competitiveness | | F3 Technological competitiveness | | F4 Organizational competitiveness | | F5 Competitiveness of external environment | | F Comprehensive competitiveness | |
|---|---|---|---|---|---|---|---|---|---|---|---|---|
| | 2004 | 2018 | 2004 | 2018 | 2004 | 2018 | 2004 | 2018 | 2004 | 2018 | 2004 | 2018 |
| Moranehen | −0.119 | −0.183 | −0.243 | −0.210 | −0.094 | −0.148 | −0.174 | −0.145 | −0.202 | −0.206 | −0.158 | −0.208 |
| E(I) | −0.143 | −0.143 | −0.143 | −0.143 | −0.143 | −0.143 | −0.143 | −0.143 | −0.143 | −0.143 | −0.143 | −0.143 |
| SD(I) | 0.069 | 0.079 | 0.080 | 0.083 | 0.068 | 0.068 | 0.079 | 0.076 | 0.082 | 0.081 | 0.076 | 0.079 |
| z | 0.345 | −0.517 | −1.244 | −0.807 | 0.720 | −0.077 | −0.399 | −0.031 | −0.723 | −0.772 | −0.197 | −0.815 |
| p-value* | 0.730 | 0.605 | 0.214 | 0.420 | 0.471 | 0.938 | 0.690 | 0.975 | 0.470 | 0.440 | 0.844 | 0.415 |

Note: * indicates significance at the 10% level

** indicates significance at the 5% level

*** indicates significance at the 1% level.

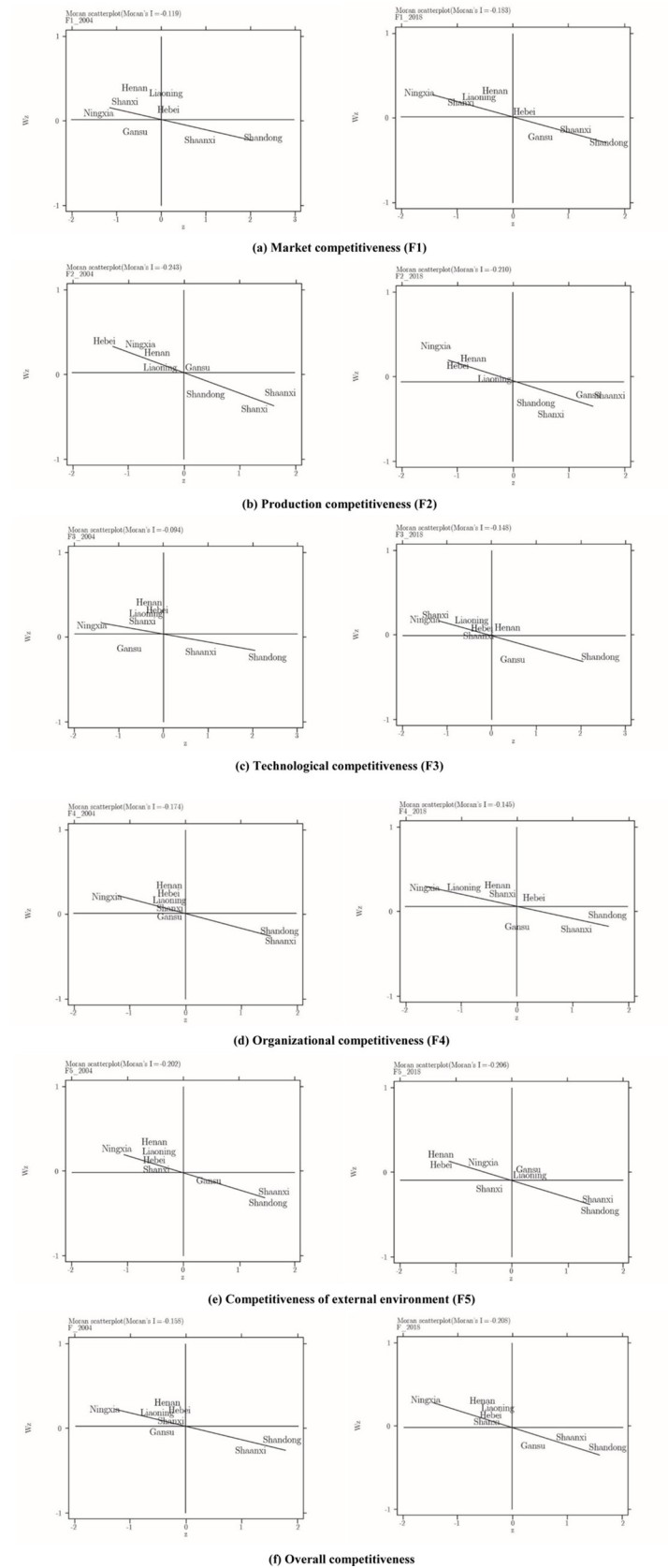

**Fig 1. Scatterplots of core variables of competitiveness in 2004 and 2018.**

It can be seen from the local autocorrelation graph that the F1 Moran index is −0.119 (Fig 1), indicating that the space of each region is negatively correlated. Hebei is in a high-value area and is surrounded by more high-value areas. Shaanxi and Shandong are in high-value areas but are surrounded by low-value areas. Shanxi, Liaoning, Henan, and Ningxia are all in low-value areas but are surrounded by high-value areas, and Gansu is located in a low-value area and is surrounded by low-value areas. Regarding the F2 index, the local Moran's I is −0.243. Hebei, Ningxia, Henan, and Liaoning are in an area of low value surrounded by high value. Gansu is in an area of high value surrounded by high value. Shandong, Shaanxi, and Shanxi are in an area of high value surrounded by low value. For F3, Moran's I is −0.094. Shandong and Shaanxi are in an area of high value surrounded by low value. Hebei, Ningxia, Henan, Liaoning, and Shanxi are surrounded by low values and high values. Hebei moves toward the positive X-axis, but its own value is still high, and its correlation degree with the surrounding areas is small. The Moran's I of F4 is −0.174. Shandong and Shaanxi are in an area where high values are surrounded by low values. Hebei, Ningxia, Henan, Liaoning, and Shanxi are in an area where low values are surrounded by high values. Gansu is in an area where high values are surrounded by high values. The Moran's I of F5 is −0.202. Shandong, Shaanxi, and Gansu are in an area where high values are surrounded by low values. All other provinces are in an area where low values are surrounded by high values.

In 2018, the Moran's I of F1 was −0.183, and its absolute value was larger than that of 2004, indicating that the local correlation of each region was enhanced. Gansu moved from an area of low value surrounded by low value to an area of high value surrounded by low value. The rest of the region did not change much compared with 2004. Hebei is in a high-value area close to the X-axis, indicating that it still belongs to the high value, but the influence of high or low value is no longer obvious. The Moran's I for F2 is −0.210, and its absolute value is smaller than the local Moran's I in 2004. Therefore, the spatial correlation was weakened, and Gansu moved from a high-value area affected by a high-value area to a high-value area affected by a low-value area. Hebei, Ningxia, Henan, and Liaoning are still in areas where low values are affected by high values, and Shandong, Shaanxi, Shanxi, and Gansu are still in areas where high values are affected by low values. The Moran's I for F3 is −0.148. Shandong and Gansu are in an area where high values are surrounded by low values. Hebei, Ningxia, Liaoning, and Shanxi are in an area where low values are surrounded by high values. Henan is in an area where high values are surrounded by high values, and Shaanxi is in an area where low values are surrounded by low values. However, when Shaanxi moves toward the edge of the negative part of the X-axis, its value is low, and it is less affected by the surroundings. The Moran's I of F4 is −0.145, which is not much different from 2004. Gansu moved from an area of low value surrounded by low value to an area of high value surrounded by low value. Hebei moved from a low-value area to a high-value area. It was originally surrounded by high value, which gradually changed into a weakened correlation with the surrounding areas. The Moran's I of F5 is −0.206, and its regional correlation is enhanced. Shanxi moved from a low-value area surrounded by high-value areas to a low-value area surrounded by low-value areas. Gansu and Liaoning changed from low-value areas to high-value areas, and their high values gradually weakened, approaching the central point.

In 2004, the Moran's I of the comprehensive index was −0.158. Shandong and Shaanxi are in an area of high value surrounded by low value, and Gansu is on the X-axis of the low-value area. It has a low value, but its correlation with the surrounding areas is weak. The remaining provinces are in low-value areas surrounded by high values. In 2018, the Moran's I for the composite index was −0.208, its absolute value greater than the 2004 value, indicating increased regional correlation. The local correlation is basically stable; the only difference is

**Table 5. Spatial correlation of competitiveness types in Chinal spatial dependence and no global spatial correlat8.**

| Competitive type | First quadrant (h-h) | | Second quadrant (l-h) | | Third quadrant (l-l) | | Fourth quadrant (h-l) | |
|---|---|---|---|---|---|---|---|---|
| | 2004 | 2018 | 2004 | 2018 | 2004 | 2018 | 2004 | 2018 |
| F1 Market competitiveness | Hebei | Hebei | Ningxia, Henan, Liaoning, Shanxi | Ningxia, Henan, Liaoning, Shanxi | Gansu | - | Shandong, Shaanxi | Shandong, Shaanxi, Gansu |
| F2 Production competitiveness | Gansu | - | Hebei, Ningxia, Henan, Liaoning | Hebei, Ningxia, Henan, Liaoning | - | - | Shandong, Shaanxi, Shanxi | Shandong, Shaanxi, Shanxi, Gansu |
| F3 Technological competitiveness | - | Henan | Hebei, Ningxia, Henan, Liaoning, Shanxi | Hebei, Ningxia, Liaoning, Shanxi | Gansu | Shaanxi | Shandong, Shaanxi | Shandong, Gansu |
| F4 Organizational competitiveness | - | Hebei | Hebei, Ningxia, Henan, Liaoning, Shanxi | Ningxia, Henan, Liaoning, Shanxi | Gansu | - | Shandong, Shaanxi | Shandong, Shaanxi, Gansu |
| F5 External environment competitiveness | - | Gansu, Liaoning | Hebei, Ningxia, Henan, Liaoning, Shanxi | Hebei, Ningxia, Henan | - | Shanxi | Shandong, Shaanxi, Gansu | Shandong, Shaanxi |
| F Comprehensive competitiveness | - | - | Hebei, Ningxia, Henan, Liaoning, Shanxi | Hebei, Ningxia, Henan, Liaoning, Shanxi | Gansu | - | Shandong, Shaanxi | Shandong, Shaanxi, Gansu |

that Gansu Province moved from a low-value, low-value-encircled area to a high-value area, but its correlation with the surrounding area was enhanced.

Table 5 shows that China's apple industry has no global spatial dependence and no global spatial correlation. During the study period, the comprehensive competitiveness of China's apple industry shows a spatial distribution of high-competitiveness regions, represented by Gansu, Shaanxi, and Shandong, surrounded by low-competitiveness regions. Hebei, Henan, Ningxia, Shanxi, and other regions show the spatial distribution characteristics of low-competitiveness regions surrounded by high-competitiveness regions. Gansu has moved from being surrounded by low-competitiveness regions to a region with high competitiveness surrounded by low-competitiveness regions. Its correlation with surrounding regions has been enhanced, and its own competitiveness has also been enhanced. Shandong and Shaanxi are still surrounded by regions with high competitiveness and low competitiveness, indicating that the competitiveness of these two regions is high, and the region has stability. The increase in the apple industry competitiveness of Shandong and Shaanxi, as the representative regions, slows down and gradually stabilizes, and the correlation with surrounding regions is enhanced.

Moreover, the intensification degree of China's apple industry is not high, and there is no global spatial correlation. With the continuous improvement of the competitiveness structure of the apple industry, the spatial correlation of the five competitiveness types in the eight main production areas shows constant change during the study period, with local regional correlation gradually stabilizing by 2018.

## Discussion and policy implications

Based on data for eight Chinese provinces in 2004 and 2018, this study uses PCA and spatial auto-correlation analysis to analyze the competitiveness of the apple industry in terms of the aspects of market, production, technology, organization, and environment. To date, there are no studies reporting sustainable competitiveness of apple industry based on six dimensions. This paper explains the spatial differentiation characteristics of apple industry competitiveness in China and makes some theoretical contributions to promote the sustainable development of apple industry.

### Discussion

The range of competitiveness varies in different regions. First, Shanxi and Gansu show large increases. During the study period, the ranking of Shanxi Province improved greatly. A

possible reason is that Shanxi Province is located in the east of the Loess plateau, in the apple-planting industrial belt, and over the years, Shanxi's apple production has been restructured, based on advanced technology and equipment; this finding is consistent with previous studies [62]—that is, the adjustment of agricultural industrial structure based on scientific and technological progress can effectively promote the extension of agricultural industrial chain [63]. In Gansu, meanwhile, the government has supported the local apple industry, which is one reason for its relatively stable development [64]. Second, Shandong, Ningxia, and Shaanxi show stable performance. Shandong is a traditional dominant producing area in China, Shaanxi is a large apple-growing province, and Ningxia has become a region suitable for apple production as the country's apple industry has developed from east to west in recent years. Third, Liaoning, Hebei, and Henan have fallen in the rankings. The possible explanation is that the lack of product research and development and insufficient publicity in these three main apple-producing areas have gradually weakened market competitiveness [65,66]. Zhang et al. [67] pointed out that the higher the level of agricultural scientific and technological innovation, the more significant the agricultural economic growth and thus the sustainable development of agriculture.

The overall competitiveness of China's apple industry is insufficient. The results obtained using PCA and spatial autocorrelation are consistent with the objective situation in China. About three-fifths of the provinces score below 0 in competitiveness; the competitiveness level of the apple industry is generally low, and the gaps are large [68]. In addition, the level of China's apple production scale economy is significant, and there are diminishing returns to scale. First, Shandong, Shaanxi, and Gansu have strong competitiveness. These regions should, therefore, form a regional advantage band with sustainable competitiveness to improve the overall sustainable competitiveness of the apple industry. Second, apple-related professional organizations should cooperate to integrate land resources and form a batch of villages, large growers, and industrial bases. Third, professional cooperatives and leading enterprises can coordinate apple storage, transportation, insurance, processing, marketing, and other industrial chain links and promote cold chain logistics and processing to form sustainable competitiveness in the industry.

## Policy implications

Based on the above conclusions, the following suggestions are made for the sustainable development of the apple industry.

(1) Improve the degree of industrial intensification according to different regional development levels. Shandong should take advantage of its long-term apple-planting advantages, constantly update its technology, achieve industrial optimization, and cultivate demonstration parks for apple industry development. Under the premise of rapid scientific and technological development, Gansu and Ningxia can create new growth points for the apple industry. The area of the apple industry should be increased without competing with other fruits for resources. Development plans should aim to alleviate poverty, transform arid areas, make efficient use of residual orchard areas, and form a large pattern of interactive and coordinated apple industry development across the country. (2) Improve the development of science and technology and enhance the sustainable competitiveness of the apple industry. First, to promote continuous progress in competitiveness, multiple varieties of apple should be produced. Second, with the help of enterprises and the government, further R&D can enhance the apple market. Regulations should be formulated to standardize the process, from apple production to selling, thus improving the sustainable competitiveness of the industry. (3) Change the production mode and develop new apple subjects. Excellent orchard farmers should be cultivated.

Moreover, cooperation between farmers and enterprises and between the market and production should be promoted; apple products should be supported in meeting market demands. (4) Increase the share of apples in the fruit market through publicity. For example, the medicinal value of apples can be used to promote them and stimulate consumption with the help of new media. Apple-processing enterprises should also strengthen their technology and capital investment and strive to produce products that cater to the market (e.g., canned apples and applesauce). They should, moreover, strengthen publicity and stimulate the improvement of the industry's market competitiveness. (5) Improve the policy support system and enhance competitiveness. To support apple industry development, the proportion of fruit farmers' planting input and income needs to be balanced. Therefore, the government should provide more policy support to each link of the apple industry chain. First, subsidies should be provided for apple production, which can improve fruit farmers' enthusiasm to a certain extent. Second, the price mechanism should be established and improved. Third, enterprises and fruit farmers should be encouraged to develop apple-processing projects. Fourth, relevant insurance policies should be provided for fruit farmers to improve the apple industry planting system.

## Conclusion and limitations

### Conclusions

First, during the study period, the competitiveness of China's apple industry gradually increased, and the growth rate was large, although it gradually slowed down. In the eight main apple-producing areas, the competitiveness gap is obvious. Shandong, Shaanxi, and Gansu are relatively strong, while Henan, Hebei, and Ningxia are at a low level of competitiveness. Shanxi and Gansu are constantly climbing in the rankings, and their market shares are constantly increasing. In recent years, the apple industry's competitiveness has steadily improved as a result of investment in scientific production, and the industry presents a good overall development trend.

Second, there is no global spatial dependence in China's apple industry; rather, there is local correlation and gradually stability. Because of natural environmental factors, the degree of industrial intensification of China's apple industry is not high, and spatial aggregation did not form during the study period. In recent years, with the steady improvement of the apple industry's competitive structure, the spatial correlation of the five types of competitiveness among the eight production areas has also been changing, with obvious changes influenced by surrounding regions. Industry competitiveness in Shandong, Shaanxi, Gansu, and other regions is relatively coordinated while it is less so in Ningxia, Henan, and other regions.

### Limitations

Although we carried out comprehensive research, there were limitations to this study. The general situation of eight major apple-producing regions in China was used to construct the evaluation index system of the agricultural economic competitiveness in this paper. Although the levels are clear and the coverage is wide, due to the limited reference materials, some index data could not be obtained. A significant amount of data was missing, which led to a small number of third-level indicators in agricultural innovation. Therefore, the evaluation index system of the apple industry competitiveness designed in this paper needs to be further improved. Further research is needed on the ecological environment of the apple industry in China, which is more conducive to exploring the competitiveness and sustainable development of the apple industry in China.

## Supporting information

**S1 File.**
(XLSX)

## Acknowledgments

We are indebted to the anonymous reviewers and editors.

## Author Contributions

**Conceptualization:** Yu Sun.

**Data curation:** Yu Sun, Yonghua Lu, Mingyue Li.

**Formal analysis:** Yu Sun.

**Funding acquisition:** Yu Sun.

**Investigation:** Yu Sun.

**Methodology:** Yu Sun, Ye Deng.

**Writing – original draft:** Ye Deng.

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
