## [Decision Letter · Decision Letter 0]

3 Feb 2022

PONE-D-22-00683Competitiveness and Sustainable Development of China’s Apple IndustryPLOS ONE

Dear Dr. Lu,

Thank you for submitting your manuscript to PLOS ONE. After careful consideration, we feel that it has merit but does not fully meet PLOS ONE’s publication criteria as it currently stands. Therefore, we invite you to submit a revised version of the manuscript that addresses the points raised during the review process.

I like your study in general but there are some small points you have to correct. Especially reasoning of your study is not complete, please see my comments and reviewer comments. I think you need to add that funders had not had any effect on design of your study and your conclusions

Please submit your revised manuscript by 3 March 2022 If you will need more time than this to complete your revisions, please reply to this message or contact the journal office at plosone@plos.org. Please include the following items when submitting your revised manuscript:A rebuttal letter that responds to each point raised by the academic editor and reviewer(s). You should upload this letter as a separate file labeled 'Response to Reviewers'.A marked-up copy of your manuscript that highlights changes made to the original version. You should upload this as a separate file labeled 'Revised Manuscript with Track Changes'.An unmarked version of your revised paper without tracked changes. You should upload this as a separate file labeled 'Manuscript'.If applicable, we recommend that you deposit your laboratory protocols in protocols.io to enhance the reproducibility of your results. Protocols.io assigns your protocol its own identifier (DOI) so that it can be cited independently in the future. For instructions see: https://journals.plos.org/plosone/s/submission-guidelines#loc-laboratory-protocols. Additionally, PLOS ONE offers an option for publishing peer-reviewed Lab Protocol articles, which describe protocols hosted on protocols.io. Read more information on sharing protocols at https://plos.org/protocols?utm_medium=editorial-email&utm_source=authorletters&utm_campaign=protocols.

We look forward to receiving your revised manuscript.

Kind regards,

Ahmet Uludag, Ph.D.

Academic Editor

PLOS ONE

Journal Requirements:

"This research was funded by Shandong Province Modern Agricultural Industrial Technology System (SDAIT-01-13) and the Project of Shandong Provincial Department of Education (M2020111) and Qingdao Social Science Planning Project: Research on the Interactive Relationship between Agricultural Informatization and Agricultural Economic Growth in Qingdao under the Background of Digital Rural Strategy（QDSKL2101221）."

7. PLOS requires an ORCID iD for the corresponding author in Editorial Manager on papers submitted after December 6th, 2016. Please ensure that you have an ORCID iD and that it is validated in Editorial Manager. To do this, go to ‘Update my Information’ (in the upper left-hand corner of the main menu), and click on the Fetch/Validate link next to the ORCID field. This will take you to the ORCID site and allow you to create a new iD or authenticate a pre-existing iD in Editorial Manager. Please see the following video for instructions on linking an ORCID iD to your Editorial Manager account: https://www.youtube.com/watch?v=_xcclfuvtxQ

Additional Editor Comments:

Could you please complete following part in the submission "Did the sponsors or funders play any role in the study design, data collection and analysis, decision to publish, or preparation of the manuscript?

NO - Include this sentence at the end of your statement: The funders had no role in study design, data collection and analysis, decision to publish, or preparation of the manuscript.

YES - Specify the role(s) played.

Both reviewers have shown weak points of your manuscript. Especially you need to complete reasoning why you did this research, why you have selected theses parts of china and these two years.

Reviewers' comments:

Reviewer's Responses to Questions

**Comments to the Author**

1. Is the manuscript technically sound, and do the data support the conclusions?

Reviewer #1: Yes

Reviewer #2: Yes

2. Has the statistical analysis been performed appropriately and rigorously? 

Reviewer #1: Yes

Reviewer #2: Yes

3. Have the authors made all data underlying the findings in their manuscript fully available?

Reviewer #1: Yes

Reviewer #2: No

4. Is the manuscript presented in an intelligible fashion and written in standard English?

Reviewer #1: Yes

Reviewer #2: Yes

5. Review Comments to the Author

Reviewer #1: The peer-reviewed article deals with the assessment the competitiveness of China’s apple industry and the factors affecting it. Using 2004 and 2018 data for eight Chinese provinces, principal component analysis and spatial autocorrelation are used to examine competitiveness in terms of five aspects: market, production, technology, organization, and environment. As a result of the conducted reseach, the author draws conclusions about improving scientific production, reasonable capital investment etc. In general, the peer-reviewed article meets the requirements of the journal, but the reviewer has a few minor remarks:

1) The title of the article - is "Competitiveness and Sustainable Development of China’s Apple Industry". What does the author mean by the "Sustainable development" of the industry? The text of the papere is about the sustainable of competitiveness.

2) It is not clear from the text of the article why the author chose 2004 and 2018 for the study? And will the proposed methods of assessing competitiveness for other periods of time be relevant?

3) I support the author's opinion that, of course, 8 regions are not enough to conduct a study (p. 19). Therefore, the evaluation index system of the apple industry competitiveness designed in this paper needs to be further improved.

Reviewer #2: Comments to authors

The manuscript is interesting as it helps to analyze and promote the sustainable competitiveness of China’s apple industry. However, some comments need to be addressed to improve the manuscript to meet the publication standard.

1. Manuscript needs proofreading. There are a lot grammatical errors, e.g., line 147-148, line 153-155, product marketing refers….,..

2. The paragraph line 189 is very short.

3. The author should justify why inverse distance weight matrix was used over the binary adjacency matrix or other methods

4. Table 2 looks mixed up, the author should work on the table 2 and make it clear

5. Where is section 4.2? there are section 4.1 and section 4.3, please check.

6. The discussion should be elaborated by comparing the current study with the results from the previous studies

7. The conclusion should be brief and concise, there should be no numbering in the conclusion.

6. PLOS authors have the option to publish the peer review history of their article (what does this mean?). If published, this will include your full peer review and any attached files.

Reviewer #1: **Yes: **Olena Bohdaniuk, PhD, Associate Professor

Reviewer #2: **Yes: **Ocident Bongomin

---

## [Author Response · Author response to Decision Letter 0]

14 Apr 2022

Reviewer #1:

（1） The title of the article - is "Competitiveness and Sustainable Development of China’s Apple Industry". What does the author mean by the "Sustainable development" of the industry? The text of the papere is about the sustainable of competitiveness.

Thank you very much for your valuable advice.This is explained in the introduction, lines 28-31. Sustainable competitiveness as understood in this paper is composed of some factors, and then it can bring about the sustainable development of apple industry. thank you The original text reads as follows:In the context of industry, sustainable competitiveness refers to the ability of an industry in one area, compared with the same type industry in other areas, to accrue and optimize resources and production factors to meet consumers’ needs and to survive and develop.

Thank you very much.

（2）It is not clear from the text of the article why the author chose 2004 and 2018 for the study? And will the proposed methods of assessing competitiveness for other periods of time be relevant?

Thank you very much for your valuable advice.This paper studies the sustainable competitiveness of apple industry. Firstly, the stability and representativeness of data research results should be ensured. Secondly, I consulted previous literatures and considered the research methods of previous scholars. Third, 2004 is the beginning of the study period, and 2018 is the end of a period. Therefore, this paper takes 2004 and 2018 as representative years to investigate the sustainable competitiveness of eight major apple producing areas.

（3）I support the author's opinion that, of course, 8 regions are not enough to conduct a study (p. 19). Therefore, the evaluation index system of the apple industry competitiveness designed in this paper needs to be further improved.

In the future, I will have the opportunity to study Apple industry in China through practical research. Considering the availability of data, the principle of representativeness and the experience of previous studies, eight major producing areas were selected as research regions. On the one hand, apple cultivation and production in China are mainly concentrated in these eight regions, so it is representative to select these regions for research. On the other hand, the data in China's agricultural statistics Yearbook and agricultural product Data compilation are only from these eight major apple producing areas, which are authoritative and reliable. thank you.

Reviewer #2: 

Thank you very much for your valuable advice. 

1. Manuscript needs proofreading. There are a lot grammatical errors, e.g., line 147-148, line 153-155, product marketing refers….,..

I recalibrated it. Thank you.

2. The paragraph line 189 is very short.

I rearranged it and put it at the end of the last paragraph.

3. The author should justify why inverse distance weight matrix was used over the binary adjacency matrix or other methods

I looked at the literature，the inverse distance weight method is a spatial distribution method that considers regional correlation among various factors and has been widely used in data processing due to its simple principle and accurate results. The inverse distance weighting matrix is based on the assumption that space effect intensity depends on distance. Except for the main apple-producing units, the effect size of all other evaluation units decreases with increasing distance, which is consistent with the first law of geography stating that "anything is always correlated, and things that are close are more correlated than things that are far away." Meanwhile, a non-sparse matrix can be transformed into a sparse matrix to solve the problem of long running time caused by excessive number of evaluations. Therefore, the inverse distance weight matrix is used as the basis for spatial autocorrelation analysis.

4. Table 2 looks mixed up, the author should work on the table 2 and make it clear.

The horizontal space of the table is limited, so I have changed the format according to the situation, please have a look.Thank you.

5. Where is section 4.2? there are section 4.1 and section 4.3, please check.

Section 4.2 is on page 9 of the paper.Thank you.

6. The discussion should be elaborated by comparing the current study with the results from the previous studies.

I changed the title of part 5 to discussion and policy implications, and compared the results of the current research with those of the previous research you said, added some literature and marked them in the paper.

7. The conclusion should be brief and concise, there should be no numbering in the conclusion.

According to your modification requirements, I have removed the serial number of the conclusion and deleted a few things.

---

## [Decision Letter · Decision Letter 1]

1 May 2022

Competitiveness and Sustainable Development of China’s Apple Industry

PONE-D-22-00683R1

Dear Dr. Lu,

We’re pleased to inform you that your manuscript has been judged scientifically suitable for publication and will be formally accepted for publication once it meets all outstanding technical requirements.

Kind regards,

Ahmet Uludag, Ph.D.

Academic Editor

PLOS ONE

Additional Editor Comments (optional):

Thank you for your precise work after comments.

Reviewers' comments:

Reviewer's Responses to Questions

**Comments to the Author**

1. If the authors have adequately addressed your comments raised in a previous round of review and you feel that this manuscript is now acceptable for publication, you may indicate that here to bypass the “Comments to the Author” section, enter your conflict of interest statement in the “Confidential to Editor” section, and submit your "Accept" recommendation.

Reviewer #1: All comments have been addressed

Reviewer #2: All comments have been addressed

2. Is the manuscript technically sound, and do the data support the conclusions?

Reviewer #1: Yes

Reviewer #2: Yes

3. Has the statistical analysis been performed appropriately and rigorously? 

Reviewer #1: Yes

Reviewer #2: Yes

4. Have the authors made all data underlying the findings in their manuscript fully available?

Reviewer #1: Yes

Reviewer #2: No

5. Is the manuscript presented in an intelligible fashion and written in standard English?

Reviewer #1: Yes

Reviewer #2: Yes

6. Review Comments to the Author

Reviewer #1: The author took into account the comments made in the previous review. The answers to the reviewer's comments were clear, therefore, I consider, that the peer-reviewed article may be allowed to be published.

Reviewer #2: The author should do thorough proofreading on manuscript and follow the journal manuscript template before submitting it for publication.

Please check, The manuscript has two contradicting headings: Conflicts of interest and Competing interest.

7. PLOS authors have the option to publish the peer review history of their article (what does this mean?). If published, this will include your full peer review and any attached files.

Reviewer #1: **Yes: **Olena Bohdaniuk

Reviewer #2: **Yes: **Ocident Bongomin

---

## [Editor Report · Acceptance letter]

19 May 2022

PONE-D-22-00683R1 

Competitiveness and Sustainable Development of China’s Apple Industry 

Dear Dr. Lu:

I'm pleased to inform you that your manuscript has been deemed suitable for publication in PLOS ONE. Congratulations! Your manuscript is now with our production department. 

Kind regards, 

on behalf of

Dr. Ahmet Uludag 

Academic Editor

PLOS ONE